# Hybrid Polydopamine/Ag Shell-Encapsulated Magnetic Fe_3_O_4_ Nanosphere with High Antibacterial Activity

**DOI:** 10.3390/ma13173872

**Published:** 2020-09-02

**Authors:** Qunling Fang, Kezhu Xu, Jianfeng Zhang, Qingshan Xiong, Jinyu Duan, Shouhu Xuan

**Affiliations:** 1Key Laboratory of Metabolism and Regulation for Major Diseases of Anhui Higher Education Institutes, School of Food and Biological Engineering, Hefei University of Technology, Hefei 230009, China; 2018111342@mail.hfut.edu.cn (K.X.); 2016170983@mail.hfut.edu.cn (J.Z.); 2019171293@mail.hfut.edu.cn (Q.X.); 2017170991@mail.hfut.edu.cn (J.D.); 2CAS Key Laboratory of Mechanical Behavior and Design of Materials, Department of Modern Mechanics, University of Science and Technology of China, Hefei 230027, China

**Keywords:** core-shell, magnetic, Ag, antibacterial, polydopamine

## Abstract

The bacteria, which usually contaminate water environment, often cause terrible infectious diseases thus seriously threaten people’s health. To meet the increasing requirement of the public health care, an easily separable nanomaterial with sustainable anti-bacteria performance is required. This work reports a Fe_3_O_4_@PDA/Ag/PDA core-shell nanosphere in which the Ag nanocrystals immobilized on the magnetic carrier are protected by an external polydopamine (PDA) layer. The magnetic hybrid nanospheres are constructed by a tunable coating method and the particle parameters can be effectively controlled by the experimental condition. The antibacterial potential of the nanospheres is evaluable by using the *Staphylococcus aureus* and *Escherichia coli* as the models. The results indicate the Fe_3_O_4_@PDA/Ag/PDA core-shell nanospheres have a high antibacterial performance by measuring the minimum inhibitory concentration and the minimum bactericidal concentration. Finally, the product is expected to have a sustainable activity because the protecting PDA layer reduce the releasing rate of the Ag^+^ ions and the materials can be magnetically recovered from the media after the disinfection procedure.

## 1. Introduction

The bacteria often cause terrible infectious diseases thus seriously threaten public health. To reduce the infection, the effective antibiotics are usually used to save people’s lives [1]. However, with the development of the bacterial resistance, the efficacy of the antibiotics is keeping weakened [2]. Recently, the nano-antibacterial agents have attracted tremendous attention due to their negligible resistance, high selectivity, low invasiveness, and weak systemic toxicity [3,4,5]. For centuries, Ag has been proven as an important antibacterial agent which is widely applied in water purifying and treating infections, burns, and ulcers [6,7,8]. The studies indicated that the smaller sized Ag nanoparticles often exhibited stronger antibacterial activity and improved biocompatibility [9]. Therefore, much effort has been conducted to develop Ag-based nano-antibacterial agent for antibacterial drugs, food packaging, medical bandages, and clothing [10,11,12].

Uniform dispersing, controllable releasing, and easy separation are the three key factors considered in the nano-antibacterial agents. Because of the high surface energy, the Ag nanoparticles are easy to be aggregated in colloidal solution and subsequently reduce the antimicrobial activity. To this case, many Ag-nanocomposites have been fabricated to improve the dispersity by immobilizing the Ag nanocrystals on nanocarriers or being encapsulated within the porous templates, respectively [13]. To extend the effectiveness, the incorporation of Ag nano-antibacterial agents into a protection matrix is also favorable to enhance the sustainability [14]. Since the Ag antibacterial nanocomposites are applied in human contacting areas, an environment friendly synthesis process is preferred and a final bio-compatible surface is of growing requirement. It is reported that many biocompatible polymers, especially the dopamine, can be utilized in synthesizing noble metal nanoparticles as both reducing agents and protection layer [15]. Interestingly, the dopamine can firmly adhere on diverse substrates via strong intermolecular interactions and final polydopamine (PDA) presents a meso-porous characteristic [16,17,18]. As a result, the Ag antibacterial nanocomposite with a polydopamine protecting shell will not only preserve the Ag nanoparticles from quick releasing but also reduce the surface toxicity.

Moreover, high concentration of Ag nano-antibacterial agents is also toxic to mammalian cells, although they exhibited outstanding antibacterial performance. The efficient removing of the Ag antibacterial nanocomposites from specific targets as much as possible after the bacteria has been killed is necessary. Magnetic field removing has been widely applied in separation, nanocatalysis, absorption and so on owing to the easy remote conduction [19,20,21,22]. The previous reports also proved that the magnetic nano-antibacterial agents exhibited high activity on antibacterial and showed interesting recoverable and reusable characteristic [23]. With the rapid developing requirements of modern public health care, a sustainable Ag nano-antibacterial agent with green preparation, bi-compatible surface, thin protecting layer, and easy separation, is still needed.

Here, a magnetically separable nano-antibacterial agent Fe_3_O_4_@PDA/Ag/PDA with a distinctive core/multi-shell nanostructure is designed and synthesized. The nanostructure of the magnetic hybrid nanospheres can be effectively controlled by tuning the experimental condition. The antibacterial potential of the nanospheres on the *Staphylococcus aureus* and *Escherichia coli* is evaluated and the results indicate that the Fe_3_O_4_@PDA/Ag/PDA core-shell nanospheres have a high antibacterial performance. The protective PDA shell not only prevents the aggregation of Ag nanoparticles but also exhibited a hydrophilic surface. Finally, the product is expected to have a sustainable activity because the protecting PDA layer reduce the releasing rate of the Ag^+^ ions and the materials can be recovered from the media after the disinfection procedure.

## 2. Experimental Section

### 2.1. Materials

Iron(III) chloride hexahydrate (FeCl_3_·6H_2_O), ethylene glycol, sodium acetate anhydrous, polyacrylic acid (PAA), ethanol (EtOH), trihydroxymethyl aminomethane (Tris), hydrochloric acid, sodium citrate (C_6_H_5_O_7_·2H_2_O), silver nitrate, (3-aminopropyl)-triethoxysilane (APTES, C_9_H_23_NO_3_Si), 3-hydroxytyrosine hydrochloride (DA-HCl) were bought from Aladdin Reagent (Shanghai, China). All chemicals were used as received without any further purification. The water used in the experiment is ultrapure water.

### 2.2. Preparation of Fe_3_O_4_@PDA Core Shell Nanospheres

First, magnetic Fe_3_O_4_ nanoparticles were synthesized by a solvothermal reaction method [24]. Typically, FeCl_3_·6H_2_O (1.08 g) and NaAc (4.0 g) were dissolved in 40 mL of ethylene glycol solution. Then, 0.1 g PAA was added and dispersed by ultrasonication for 20 min. The resulting mixture was poured into a polytetrafluoroethylene lining and transferred to a high-pressure reactor in a 200 °C oven for 12 h. After the reaction, the product was washed with deionized water multiple times, and finally placed in a vacuum drying oven at 45 °C overnight. To synthesize Fe_3_O_4_@PDA core shell nanoparticles, the Fe_3_O_4_ (20 mg) nanoparticles were first dispersed in absolute ethanol (60 mL) and then DA-HCl (6.6 mmol/L 60 mL) containing Tris buffer (pH = 8.5) was added. After reacting under ultrasonication for 3 h, the product was collected from the solution using magnetic separation. The samples were washed with deionized water multiple times and then transferred to a 45 °C vacuum drying oven for 12 h.

### 2.3. Preparation of Fe_3_O_4_@PDA/Ag/PDA Core Shell Nanospheres

The Ag nanoparticles were immobilized on the Fe_3_O_4_@PDA nanospheres via a simple reduction method [25]. First, Fe_3_O_4_@PDA (20 mg) was added to a mixed solution of ethanol (16 mL) and water (4 mL). The solution was dispersed evenly by ultrasonication for 10 min and AgNO_3_ (2 mL/6 mg) solution was added. 20 min later, trisodium citrate solution (2 mL/8 mg) was introduced. After further sonication for 10 min, the reaction was transferred to a 100 °C oil bath and refluxed for 30 min. Then, the resulting product was cooled naturally and washed with ethanol. The Fe_3_O_4_@PDA/Ag (20 mg) nanospheres were further added to ethanol (60 mL) to form a uniform dispersion. Subsequently, Tris buffer (pH = 8.5) containing DA-HCl (6.6 mmol/L 60 mL) was added. After sonication for 3 h, the powder was collected from the solution using magnetic separation and washed three times with deionized water and ethanol. At last, the final Fe_3_O_4_@PDA/Ag/PDA nanospheres were vacuum dried at 45 °C for 12 h.

### 2.4. Antibacterial Activity of Fe_3_O_4_@PDA/Ag/PDA Core Shell Nanospheres

In order to determine the antibacterial activity of the sample, *Escherichia coli* (Gram-negative bacteria, China General Microbiological Culture Collection Center, Beijing, China) and *Staphylococcus aureus* (Gram-positive bacteria, China General Microbiological Culture Collection Center, Beijing, China) were selected for antibacterial experiments [26]. (1) Measure the MIC (minimum inhibitory concentration): the fresh LB medium (Luria-Bertani medium) was divided into several glass test tubes, 5 mL per tube, and used after high temperature sterilization. Then, a certain concentration of the test sample was added to the test tube, the concentration increased according to the gradient, and a blank control group was set. Finally, 50 µL of *Escherichia coli* or *Staphylococcus aureus* bacterial solution was added into each test tube. Then, the tubes were placed in a shaker at 37 °C for 12–16 h and the growth of the bacteria in the test tube was observed to determine the MIC of the sample. (2) Determination of the bacterial growth curve containing the sample: fresh LB medium was configured and sterilized for later testing; 1/2 MIC and MIC concentration test samples to each group of media. The absorbance was measured at OD600 (the absorbance of the solution at 600 nm wavelength) every 2 h and a growth curve was drawn. (3) Plate experiment: the fresh LB agar medium was configured and poured on the plate after high temperature sterilization. The test samples with concentration of 1/2MIC, 1/4MIC, and MIC were added to the LB medium, respectively. Then, the two bacterial solutions were added and the cells were cultured at 37 °C for 12 h. After absorbing a certain amount of cultured liquid, it was spread on the prepared plate and further placed in a 37 °C incubator for 24 h to observe the growth of the colony. Here, a blank control was set with the same process.

### 2.5. Characterization

The inner nanostructure of the nanocomposite was studied by JEM-2100F field emission transmission electron microscope (FE-TEM, JEOL, Tokyo, Japan) operating at an acceleration voltage of 200 kV. The X-ray diffraction pattern (XRD) of the product was measured with a DMax-γA rotating positive X-ray powder diffractometer from Rigaku, Tokyo, Japan. The infrared spectroscopy (FTIR, Thermo Nicolet, Waltham, MA, USA) was tested using a TENSOR 27 Fourier transform infrared spectrometer with a spectral test wavenumber range of 4000–400 cm^−1^. The thermogravimetric analysis (TG, TA Instruments, New Castle, DE, USA) of the sample was completed with a DTG-60H thermogravimetry instrument in an air atmosphere at a heating rate of 10 °C/min from room temperature to 700 °C. The X-ray photoelectron spectroscopy (XPS, Thermo Electron, Waltham, MA, USA) was tested with an ESCALAB 250 photoelectron spectrometer. The hysteresis loop was recorded on the VSM (Value Stream Mapping) at room temperature with the Hysteresis Measurement of Soft and Hard Magnetic Materials (HyMDC Metis, Leuven, Belgium). The UV-vis (Ultraviolet Visible Spectrophotometer) spectrum was recorded on a TU-1901 ultraviolet spectrophotometer (Shimadzu, Kyoto, Japan). An Optia 7300DV inductively coupled plasma atomic emission spectrometer (ICP-AES, PerkinElmer, Waltham, MA, USA) was used to analyze the Ag content in the samples.

## 3. Results and Discussion

Figure 1a–c show typical TEM images of the final Fe_3_O_4_@PDA/Ag/PDA nanospheres. It can be observed that all particles are dispersed on the copper plate with an average size of about 360 nm (Figure 1a). Interestingly, the product has a core-shell structure, with a large core and a discontinuous nanocrystalline aggregation shell and a continuous PDA shell (Figure 1b). In order to further study its internal nanostructure, a higher magnification TEM image was carried out (Figure 1c). The different colors indicate the core/shell nature of the nanosphere, where the outermost off-white layer is the PDA layer, the discontinuous black nanoparticles are Ag nanocrystals, and the inner black core is Fe_3_O_4_. Between the Ag shell and the Fe_3_O_4_ core, we can see that there is an off-white intermediate zone, which is the intermediate layer PDA. This result proves that we have successfully synthesized a nanocomposite with Fe_3_O_4_ as the core and two PDA layers with an Ag layer as the outer shell. From the EDS spectrum, Fe, O, C, and Ag elements are clearly observed [27], which further proves that the four-layer structure Fe_3_O_4_@PDA/Ag/PDA is successfully synthesized. The Cu signal appeared because the copper mesh was selected as the carrier of the sample when the transmission electron microscope was taken. When scanning the sample, the copper mesh and the sample were regarded as a whole, and the Cu signal in the EDS spectrum came from the copper mesh.

As shown in Scheme 1 is the schematic formation procedure of the Fe_3_O_4_@PDA/Ag/PDA nanospheres. First, the magnetic Fe_3_O_4_ nanospheres were prepared by the well-known solvothermal method and the surface of the final nanosphere was modified by the polyacrylic acid (PAA). Then a layer of polydopamine was covered on the Fe_3_O_4_ nanosphere to form the Fe_3_O_4_@PDA core shell nanostructure. Because of the high affinity and activity, the Ag nanocrystals could be easily immobilized on the Fe_3_O_4_@PDA and the Fe_3_O_4_@PDA/Ag nanosphere was obtained. Finally, to improve the stability of the Ag nanocrystals, the Fe_3_O_4_@PDA/Ag nanosphere was further encapsulated by another PDA shell through the easy self-polymerization process. Although the preparation possessed four steps, the total time-consume is short due to the simple conduction in each step, thus this method is believed to be effective.

In this work, the transmission electron microscope was used to trace the formation process of Fe_3_O_4_@PDA/Ag/PDA nanospheres. As shown in Figure 2a is the TEM image of the Fe_3_O_4_ nanospheres. The Fe_3_O_4_ nanospheres are uniform with an average particle size of about 300 nm. Because of the PAA-modified versatility surface, they can be well dispersed in water and ethanol. Because PDA has strong adhesion to both inorganic and organic materials, a PDA coating can be very easily covered on the surface of the magnetic Fe_3_O_4_ to form Fe_3_O_4_@PDA core shell nanosphere by a simple in situ polymerization method. The as-synthesized Fe_3_O_4_@PDA has an approximately spherical morphology (Figure 2b). In comparison to the original Fe_3_O_4_ nanoparticles, the surface of Fe_3_O_4_@PDA is smoother although Fe_3_O_4_@PDA core shell nanosphere show a similar average particle size, which indicates that the surface has a uniform PDA shell. Then, the Ag nanocrystals were fixed on the surface of Fe_3_O_4_@PDA, thus some black dots appeared on the surface (Figure 2c). To prevent the aggregation and leaching of Ag nanocrystals, another PDA shell was coated on the Fe_3_O_4_@PDA/Ag nanospheres. Figure 2d shows the TEM image of Fe_3_O_4_@PDA/Ag/PDA nanosphere which has a typical core shell nanostructure. Beyond the black Ag nanocrystals, a thin pale layer is observed. Here, the particle size of Ag is in the range of tens of nanometers and the average thickness of the outer PDA shell is about 5–10 nm.

This layer-by-layer coating method is versatile for immobilized noble metal nanocrystals on the magnetic nanocarrier. First, the total core shell nanospheres can be controlled by varying the experimental conditions, such as the Fe_3_O_4_ size, PDA shell thickness, Ag coverage and etc. As shown in Figure 3a, the Fe_3_O_4_@PDA/Ag nanosphere with size of 250 nm was successfully synthesized by using the 200 nm Fe_3_O_4_ as the core template. Unfortunately, the as-formed Ag particle layer was discontinuous, since the Ag nanocrystals were difficult to be uniformly covered on the PDA layer. Therefore, the 300 nm Fe_3_O_4_ core was used to increase the attachment area of Ag nanocrystals by decreasing the curvature. Interestingly, the coverage of the Ag nanocrystals on the 360 nm Fe_3_O_4_@PDA/Ag/PDA was obviously increased. Moreover, other noble metal such as Au nanocrystals can also be easily carried on the magnetic nanospheres via a similar coating method [28]. Because the average size of the Au nanocrystals is relatively smaller than the Ag nanocrystals, the uniform coverage of the Au nanocrystals was achieved in the Fe_3_O_4_@PDA/Au/PDA nanospheres. Previously, we developed a one-step method to synthesize Fe_3_O_4_@PDA/Au/PDA nanospheres via the in situ reduction of HAuCl_4_ to Au nanocrystals and polymerization of dopamine [24]. However, the Fe_3_O_4_@PDA/Ag/PDA nanospheres cannot be obtained and the Ag nanocrystals are seriously aggregated because of the Van der Waals force and the large specific surface area. After the particles are refined to the nanometer level, a large amount of positive and negative charges are accumulated on the surface. The shape and irregularity of the particles cause the accumulation of surface charges, which makes the particles unstable and easy to agglomerate. The distance between the nanoparticles is extremely short, and the mutual Van der Waals force is much greater than their own gravity, so they tend to attract each other and reunite. Nanoparticles have a large specific surface area and high surface energy. They are in an unstable state of energy and are prone to aggregation to reach a stable state. We also attempted to attach Au and Ag nanoparticles to the PDA shell at the same time with Au doping (Figure 3b). The results prove that these particles are unevenly distributed on the surface of the PDA, which also reduced the antibacterial performance of the final material.

Figure 4a shows the XRD diffraction pattern of the Fe_3_O_4_ nanospheres. All the characteristic peaks can be determined to be (220), (311), (440), 422), (511), and (440) lattice planes [29], which can be indexed to be face-centered cubic structured magnetite (JCPDS card No. 19-629). Because PDA is an amorphous polymer, no additional diffraction peak is found in Fe_3_O_4_@PDA, and its diffraction peak is similar to Fe_3_O_4_ (Figure 4b). Moreover, this result also indicates that the crystalline phase of Fe_3_O_4_ did not change during the PDA coating process. As shown in Figure 4c, after fixing the Ag nanocrystals on the Fe_3_O_4_@PDA surface, three distinct Ag diffraction peaks (111), (200), and (220) appear at 38°, 44°, and 65° (JCPDS card number 04-0783) [30], demonstrating the successful immobilization of Ag nanocrystals. Because the PDA shell has no effect on the XRD diffraction intensity, the peak shape of the final Fe_3_O_4_@PDA/Ag/PDA nanospheres is similar to that of Fe_3_O_4_@PDA/Ag (Figure 4d).

Figure 5a is the FTIR spectrum of the Fe_3_O_4_ core, where the broad peaks at 1413 cm^−1^ and 1622 cm^−1^ are corresponded to the vibrational absorption of the COO– and O–H bonds of the PAA, which is used as the surfactant during the synthesis of Fe_3_O_4_ nanosphere. Here, the peak observed at 3440 cm^−1^ is originated from the stretching vibration of O–H. After coating a PDA shell on Fe_3_O_4_, the strong vibration absorption band (Fe–O vibration absorption) at 595 cm^−1^ gradually weakened. The PDA shell is very thin and the absorption band of the PDA is weak, so the curve of Fe_3_O_4_@PDA has no additional key absorption peaks (Figure 5b). When Ag nanoparticles are fixed to the surface of the PDA shell, the vibration absorption of the Fe–O bond at 595 cm^−1^ is further reduced (Figure 5c). Finally, the broad absorption between 1000 and 1700 cm^−1^ demonstrates the presence of PAA and PDA polymers in the Fe_3_O_4_@PDA/Ag/PDA nanospheres (Figure 5d).

The synthesis process of the sample was also tracked by testing the TG curve in an air atmosphere in the temperature range of 25~700 °C. As shown in Figure 4, the weight loss rates of each sample are 22.3%, 29.4%, 22.2%, and 28.9%. Observing the original TG curve of Fe_3_O_4_, it can be seen that its weight has decreased, because the organic groups and moisture remaining in the core of Fe_3_O_4_ disappear with increasing temperature (Figure 6a). When the surface of the Fe_3_O_4_ core is coated with the PDA shell, the weight loss rate increases by about 7.1% in comparison to Fe_3_O_4_. This is because the PDA shell will decompose and disappear in the high temperature (Figure 6b). In comparison to Fe_3_O_4_@PDA, the weight loss rate of Fe_3_O_4_@PDA/Ag reduces by 7.2% (Figure 6c) because of the presence of Ag nanoparticles. At last, the weight loss rate of Fe_3_O_4_@PDA/Ag/PDA is further increased than that of Fe_3_O_4_@PDA/Ag (Figure 6d) because of the second PDA shell.

In order to further study the internal structure of the core-shell nanocomposites, XPS is used to analyze the surface element composition of the samples [31]. The XPS spectrum of the Fe_3_O_4_ core is shown in Figure 7a and the strong signal peaks of Fe, C, and O elements can be clearly observed. Compared with Fe_3_O_4_, there is obviously an N element signal peak in the XPS spectrum of Fe_3_O_4_@PDA (Figure 7b), which must be due to the presence of the N element-containing PDA shell. At the same time, it can be seen that the Fe element signal peak sharply reduced, indicating that the PDA the shell layer is evenly coated on the surface of Fe_3_O_4_. Because of the thickness of the shell layer is thin, some Fe element signal can be detected. Once the Ag nanocrystals are immobilized on the surface of Fe_3_O_4_@PDA, a clear Ag signal peak can be observed (Figure 7c). When another layer of PDA shell was formed on the surface of Fe_3_O_4_@PDA/Ag, the relative intensity of the Ag signal peak also decreased significantly (Figure 7d). These results indicate that the magnetic Fe_3_O_4_@PDA/Ag/PDA core-shell nanocomposite has been successfully prepared.

The Fe_3_O_4_@PDA/Ag/PDA nanosphere shows a typical magnetic characteristic and the hysteresis loops of Fe_3_O_4_ and Fe_3_O_4_@PDA/Ag/PDA nanospheres were measured by VSM at room temperature (Figure 8) Since the cluster-like Fe_3_O_4_ core is composed of a large number of small secondary nanoparticles, the Fe_3_O_4_@PDA/Ag/PDA exhibits typical soft magnetic behavior. Because of the existence of the double polydopamine shell and the discontinuous Ag nanoparticle interlayer, the saturation magnetization of Fe_3_O_4_@PDA/Ag/PDA nanospheres (30.3 emu/g) is smaller than original Fe_3_O_4_ (47.1 emu/g). However, the magnetic sensitivity of Fe_3_O_4_@PDA/Ag/PDA nanospheres is large enough to be easily separated from the reaction system by magnetic separation.

Because Ag nanocrystals have a strong inhibitory effect on a variety of bacteria and show low toxicity to the human body, they are widely used in antibacterial drugs [32]. In this paper, the antibacterial properties of Fe_3_O_4_@PDA/Ag/PDA core shell nanospheres were studied using *Escherichia coli* and *Staphylococcus aureus* as bacterial models [33,34]. Figure 9 is a photograph of colony growth after culturing two bacteria in LB agar medium added with excessive amounts of different test samples for 24 h. It can be clearly observed that after adding excessive Fe_3_O_4_ and Fe_3_O_4_@PDA and cultivating for 24 h, the growth of the two bacteria is as good as that of the control group without adding any sample, which shows that the growth of the two bacteria has not been effectively inhibited. Fe_3_O_4_ and PDA cannot inhibit the growth of these two bacteria. However, after adding excessive Fe_3_O_4_@PDA/Ag/PDA and culturing for 24 h, no obvious colonies appear in the medium, which shows that Fe_3_O_4_@PDA/Ag/PDA nanospheres can inhibit the growth of these two bacteria, and the antibacterial ability must be originated from the Ag nanocrystals [35,36,37,38,39].

As shown in Table 1, it was found that the MIC (minimum inhibitory concentration) of Fe_3_O_4_@PDA/Ag/PDA nanoparticles against *Staphylococcus aureus* and *Escherichia coli* were 320 μg/mL and 200 μg/mL, respectively. The mass fraction of Ag nanocrystals in the Fe_3_O_4_@PDA/Ag/PDA sample measured by ICP-AES is about 13.1 wt%, so the MICs of Ag nanocrystals to the two bacteria are 41.9 μg/mL and 26.2 μg/mL. Figure 10a is the growth curve of Fe_3_O_4_@PDA/Ag/PDA nanospheres and *Staphylococcus aureus* in 12 h [39]. Compared with the control group, when 1/2 MIC of Fe_3_O_4_@PDA/Ag/PDA is added, the OD600 value is significantly smaller than that of the control group as the cultivation time increases, indicating that the number of *Staphylococcus aureus* survival was less than that of the control group and the relative growth has been inhibited to a certain extent. However, when MIC’s Fe_3_O_4_@PDA/Ag/PDA nanospheres were added, the growth of *Staphylococcus aureus* was more strongly inhibited with the increase of the cultivation time and *Staphylococcus aureus* did not grow at all. It can be seen that the Fe_3_O_4_@PDA/Ag/PDA nanospheres has a strong antibacterial effect on *Staphylococcus aureus*. The Fe_3_O_4_@PDA/Ag/PDA nanospheres cultured *Escherichia coli* shows a similar result (Figure 10b) [39]. The growth curve of the bacteria within 12 h can also be observed from the figure. The growth of *Escherichia coli* was also significantly inhibited, proving that the sample had a strong antibacterial effect against *Escherichia coli*.

## 4. Conclusions

In summary, we used the layer-by-layer method to synthesize Fe_3_O_4_@PDA/Ag/PDA core shell nanospheres as an antibacterial agent. The preparation mechanism is discussed and the results indicate this method is simple and controllable. Two common pathogenic bacteria, *Staphylococcus aureus* and *Escherichia coli*, were applied as models to evaluate the antibacterial properties of the prepared samples. The minimum inhibitory concentration MIC of the samples against *Staphylococcus aureus* and *Escherichia coli* were determined to be 320 μg/mL and 200 μg/mL, respectively. Because of the protective PDA outer layer, the Fe_3_O_4_@PDA/Ag/PDA core shell nanospheres with sustainable anti-bacteria performance can be expected. The results and application prospects also prove that the layer-by-layer coating method has certain application potential in the field of synthesizing core shell nanospheres.

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
