# Peer review of "Hybrid Polydopamine/Ag Shell-Encapsulated Magnetic Fe3O4 Nanosphere with High Antibacterial Activity"

_materials, 2020, doi:10.3390/ma13173872_

Round 1

Reviewer 1 Report

This work submitted by Fang et al. reported the synthesis of Fe3O4@PDA@Ag@PDA nanohybrids for antibacterial application. The experiments are generally designed well and the results support their conclusion. Therefore, I would like to recommend its publication after addressing the following minor concerns.

1) The information in Figure 1b and 1c were overlapped. Actually they had the same magnification. Please delete either of them.

2) Please clarify the presence of Cu signal in Figure 1e.

3) On line 186, the use of larger Fe3O4 means that it should decrease the curvature, rather than “increase the curvature”.

4) The electron contrast of Fe3O4 in Figure 3b decreased obviously. This contradicts their larger size, especially compared to Figure 3a. Please explain this result.

5) Some relevant polydopamine-mediated core-satellites have been reported before (e.g., dx.doi.org/10.1021/ja303037j; doi.org/10.1021/acsnano.5b01138). Please cite them properly in this manuscript to give a better background on this topic.

6) I did not find any results relating to the function of Fe3O4 core? If so, what is the motivation to introduce the magnetic core in this system? Additional experiments for confirming magnetic recyclability should be considered to make this manuscript as a whole.

Reviewer 2 Report

To meet the increasing requirement of the public health care, a magnetically separable Fe3O4@PDA/Ag/PDA core-shell nanosphere with sustainable anti-bacteria performance was synthesized.
It is it relevant and interesting. Because it has relevance in anti-bacterial nanomaterials. Although many similar approaches have been taken, the hybrid approach looked original to me. Compared with other published material, it add to the subject area of magnetically separable core-multi shell nanostructure.
The paper is well written and the text clear and easy to read and the conclusions are consistent with the evidence and arguments presented. They also address the main question posed.

Reviewer 3 Report

This manuscript reports the synthesis of Fe3O4@PDA/Ag/PDA core-shell nanospheres and preliminary studies of their  antibacterial performance. The manuscript is interesting and it could be potentially published but only after revision. The following issues must be addressed.

There is no magnetisation studies of the composites and no relevant discussion of magnetic properties have been provided. Without these studies, the composites are not fully characterised.

There is no clear discussion of the mechanism of antibacterial activity of the composites.

Reviewer 4 Report

The manuscript presented by Q. Fang and co-workers present the synthesis of core-shell nanoparticles composed by a large Fe3O4 core covered by an organic PDA layer encapsulating Ag particle. The work is interesting providing a detailed and extended characterization. However, even if in my opinion the paper could be a valuable resource for future work, I have some concerns that must be addressed prior to publication. 

-In Figure 1 there are many low-resolution TEM images do not give any extra information I would suggest delete some of these images and add a picture describing the core-shell structure of the system, similar to the one included in Figure 2. 

-On page 6 is written "However, the Fe3O4@PDA/Ag/PDA nanospheres cannot be obtained and the Ag nanocrystals are seriously aggregated due to the Van der Waals force and the large specific surface area." This sentence should be extended for better comprehension of a broad audience 

-Figure 4 shows a larger broadening of the Fe3O4 peaks than the ones of the Ag structure even if the particle size of the Ag particles is significatively smaller.  

Round 2

Reviewer 3 Report

Unfortunately the authors still did not provide  magnetisation studies, i.e. magnetisation measurements (e.g. using SQUID magnetometer) and corresponding magnetisation curves to be shown. It is not sufficient just to take a permanent magnet and show that sample can be attracted by a magnetic field for publication in research journal. Therefore, this issue is not addressed and  the manuscript still requires a major revision in this aspect.
